



# Arctic aquatic graminoid tundra responses to nutrient availability

Christian G. Andresen[1,2] and Vanessa L. Lougheed [2].

[1]Geography Department, University of Wisconsin Madison, Madison, WI, USA.

[2]Biological Sciences Department, University of Texas at El Paso, El Paso TX, USA.

Correspondence email: candresen@wisc.edu

**Abstract:** Unraveling the environmental controls influencing Arctic tundra productivity is paramount for advancing our predictive understanding of the causes and consequences of warming in tundra ecosystems and associated land-atmosphere feedbacks. This study focuses on aquatic emergent tundra plants, which dominate productivity and methane fluxes in the Arctic coastal plain of Alaska. In particular, we assessed how environmental nutrient availability influences production of biomass and greenness in the dominant aquatic tundra species: *Carex aquatilis* and *Arctophila fulva*. We sampled a total of 17 sites distributed across the Barrow Peninsula and Atqasuk, Alaska following a nutrient gradient that ranged from sites with thermokarst slumping or urban runoff to sites with relatively low nutrient inputs. Employing a multivariate analysis, we explained the relationship of soil and water nutrients to plant leaf macro- and micro-nutrients. Specifically, we identified soil phosphorus as the main limiting nutrient factor given that it was the principal driver of biomass and Normalize Difference Vegetation Index (NDVI) in both species. Plot-level spectral NDVI was a good predictor of leaf P content for both species. We found long-term increases in N, P and Ca in *C. aquatilis* based on historical leaf nutrient data from 1970s of our study area. This study highlights the importance of nutrient pools and mobilization between terrestrial-aquatic systems and their potential influence on productivity, carbon and energy balance. In addition, aquatic plant NDVI spectral responses to nutrients can serve as landscape hot-spot and hot-moment indicator of landscape biogeochemical heterogeneity associated with permafrost degradation, nutrient leaching and availability.

**Keywords:** NDVI, permafrost thaw, thermokarst, biomass, productivity, hot-spot, hot-moment

## 1. INTRODUCTION

In the Arctic, plant growth is limited by several factors including low temperatures, short growing-seasons (e.g. irradiance) and nutrients (Chapin et al., 1975; Shaver et al., 1998). Although Arctic temperatures have increased dramatically over recent decades with parallel increases in plant biomass, nutrients have been shown to be the main driver enhancing Arctic tundra productivity compared to temperature in long-term experimental treatments (Boelman et al., 2003; Johnson et al., 2000; Jónsdóttir et al., 2005; Shaver et al., 1998). Increased tundra productivity has generally been explained by warming-mediated processes including increases in nutrient availability through soil warming, heterotrophic decomposition, and nutrient release from mineralization of organic matter and permafrost thaw (Keuper et al., 2012; Natali et al., 2012; Pastick et al., 2019; Reyes and Lougheed, 2015). These factors highlight the complexity of tundra plant growth and production under a warming and changing Arctic with implications for carbon and energy budgets (McGuire et al., 2018; Oberbauer et al., 2007; Swann et al., 2010).



Unraveling the covarying climate and environmental controls influencing Arctic tundra
productivity is paramount for advancing our predictive understanding of the causes and
consequences of warming in tundra ecosystems and associated land-atmosphere feedbacks.
47          Nutrients play a key role influencing tundra plant production with complex effects on
ecosystem carbon balance. Early work by Chapin et al., (1975) and Shaver et al., (1998)
demonstrated that nutrients, particularly N and P, enhanced plant biomass and plant accumulated
nutrients in wet tundra communities. In contrast, temperature alone has shown no effect on
biomass production in long-term experimental treatments (Boelman et al., 2003; Johnson et al.,
2000; Jónsdóttir et al., 2005; Shaver et al., 1998). While nutrients drive productivity and
accumulation of new organic matter in the soil, nutrient enrichment can result in net carbon
losses by enhancing decomposition of old carbon stocks (Mack et al., 2004). These results
emphasize the importance of nutrient–carbon interactions in controlling ecosystem processes and
ecosystem C balance in arctic tundra.
57          Our study builds on previous experimental studies that examined nutrient impacts on wet
tundra (Beermann et al., 2015; Boelman et al., 2003; Lara et al., 2019; McLaren and Buckeridge,
2019; Shaver et al., 1998) by focusing on aquatic tundra, which are a relatively understudied
plant community in the Arctic. Aquatic emergent tundra plants are known to have the highest
productivity compared to terrestrial communities and contribute to a significant portion of
regional carbon sink and methane fluxes (Andresen et al., 2017; Joabsson and Christensen, 2001;
Lara et al., 2014). In recent decades, Arctic aquatic communities have increased in biomass and
cover (Andresen and Lougheed, 2015; Villarreal et al., 2012), likely attributed to an increase in
nutrient input leached from terrestrial systems through permafrost degradation and abrupt thaw
events into aquatic habitats (Reyes and Lougheed, 2015; Turetsky et al., 2020), but the impacts
of nutrients on Arctic aquatic plant communities have not been well documented in literature
(Andresen, 2014).
69          Nutrients have increased over the past 40 years in aquatic habitats (Lougheed et al., 2011)
with parallel biomass increases of aquatic graminoids (Andresen et al., 2017). This phenomenon
will likely become more pronounced as increasing temperatures in Arctic soils continue
enhancing nitrogen mineralization (Uhlířová et al., 2007; Weintraub and Schimel, 2003) as well
as permafrost degradation and nutrient leaching (Frey and McClelland, 2009; Keuper et al.,
2012; Reyes and Lougheed, 2015). With increased thaw and subsurface flow (Frampton et al.,
2013; Shiklomanov et al., 2013), these processes may provide substantial nutrient inputs to
freshwater ecosystems, however, there is increased need to assess the effects of these increased
nutrient inputs on aquatic tundra productivity.
78          Remote sensing has been used to detect and quantify plant productivity in Arctic systems
based on multispectral indices (Epstein et al., 2012; Pastick et al., 2019; Walker et al., 2012a).
Boelman et al., (2003) showed the applicability of the normalized vegetation index (NDVI) as a
tool to track spectral responses of wet sedge tundra to nutrients in fertilization and warming
experiments. Other studies employing digital repeat photography have successfully assessed
plant phenology, biomass and productivity by evaluating vegetation color with indices in the





visual spectral range (blue, green and red) (Andresen et al., 2018; Saitoh et al., 2012; Sonnentag
et al., 2012). Plant spectral responses to nutrient enrichment in aquatic communities are poorly
understood and its monitoring using remotely sense data would help monitor and quantify
potential carbon and energy feedbacks to the atmosphere at regional scales.
With current and projected warming and nutrients loading into Arctic aquatic systems, it
is important to understand nutrient impacts on aquatic emergent vegetation, and how these
changes can be detected and modeled using remote sensing methods. In this study, we sampled
tundra pond sites that followed a nutrient gradient that range from sites with thermokarst
slumping or urban runoff to sites with relatively low nutrient inputs. We aim to characterize
nutrient limitation of aquatic emergent tundra vegetation and spectral responses of this
vegetation to nutrient inputs. We focus on the influence of soil and water nutrients on plant
biomass and greenness of *Carex aquatilis* and *Arctophila. fulva*, the dominant aquatic emergent
vascular plants in the Arctic coastal plain (Andresen et al., 2018; Villarreal et al., 2012) to
answer the questions of: (i) how is aquatic tundra responding to nutrient availability? (ii) How
environmental nutrient status influence leaf nutrients in aquatic tundra? (iii) What are the
spectral responses (NDVI) of aquatic tundra to nutrient availability?
**2. METHODS**
**2.1 Study Sites**
This study was conducted in the Barrow Peninsula, Alaska, (W156$^0$, N70$^0$) near the town
of Uqtiaġvik (formerly known as Barrow). Physiographically, the area is located in the Arctic
Coastal Plain (ACP, ~60,000 km$^2$) of northern Alaska, which stretches from the western coast
along the Chukchi Sea to the Beaufort coastal Canadian border. The ACP is dominated by thick
continuous permafrost with high ground-ice content for the Arctic peaty lowland of the
peninsula. A complex mosaic of ice-wedge patterned ground landforms developed over
millennial seasonal cycles of cracking, heaving, and thawing producing its characteristic pond-
and lake-dominated landscape (Andresen and Lougheed, 2015; Jorgenson and Shur, 2007).
These aquatic habitats are hosts for aquatic graminoid tundra that grows in shallow standing
water with a depth range 5-50cm. This study focuses on 2 species: *C. aquatilis* and *A. fulva.*
These graminoids are the dominant cover in aquatic habitats, generally growing in the edge
and/or inside tundra ponds (Andresen et al., 2017; Villarreal et al., 2012) and their distribution is
in low- and sub-Arctic. Although these species have growth forms in moist and dry tundra
(Shaver et al., 1979), this study focuses on their aquatic phenotypes.
A total of seventeen tundra ponds were sampled in early August 2013 along a nutrient
gradient with long-term sources of nutrients. Sites were grouped in four categories according
their geographic location and nutrient source as: (i) enriched urban, (ii) enriched thermokarst,
(iii) reference, and (iv) southern (Figure 1, 2, Appendix 1). Enriched urban ponds were located
within the town of Utqiaġvik, AK and their source of nutrients was mainly from village runoff.
Enriched thermokarst ponds were situated within the Barrow Environmental Observatory (BEO),
and their nutrient inputs originate from permafrost slumping into ponds. Reference sites were
located across the region in the historical International Biological Program (IBP) sites and in the
BEO; but these sites do not contain evidence of continuous permafrost slumping. Southern
latitude ponds were located 100 km south of Utqiaġvik, near the town of Atqasuk, AK. We
sampled these ponds in order to expand the geographic footprint of the study and serve as
reference to Utqiaġvik area. It is important to note that while *C. aquatilis* occurs in all ponds, *A.*
*fulva* does not occur in thermokarst ponds nor in IBP-C and WL02 ponds (Appendix 1).
Figure 1. Map of Utqiaġvik sites sampled in this study. For site details including southern sites
see Appendix 1. Imagery © 2008 DigitalGlobe, Inc.

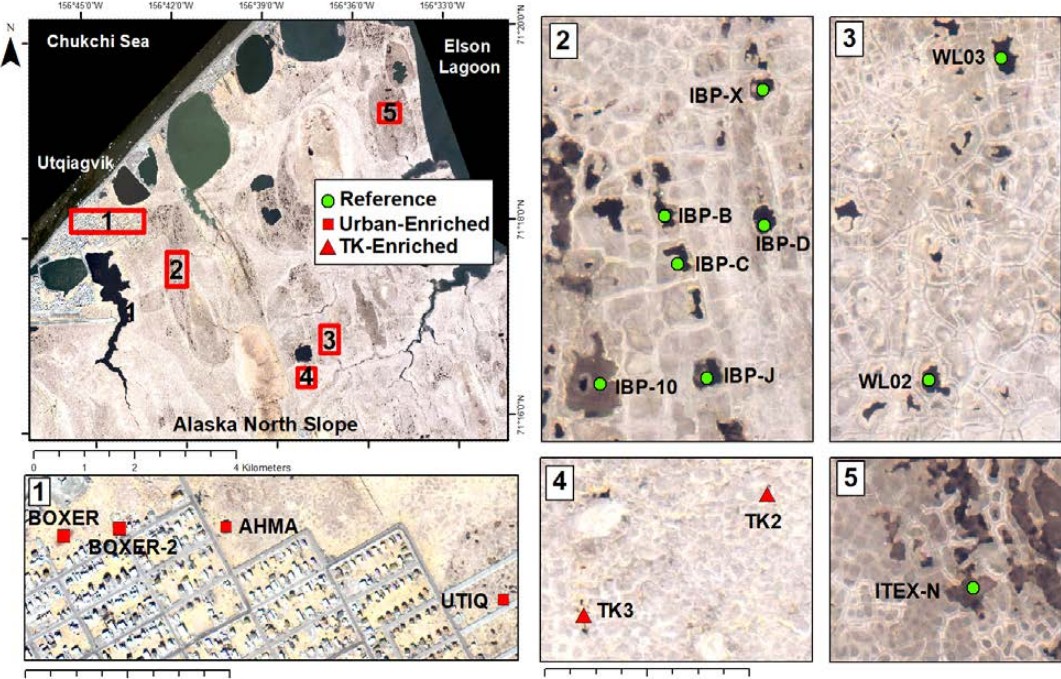




Figure 2. Aerial view the Arctic coastal plain near Utqiaġvik, AK, and examples of sites sampled
in this study. Images indicate site name (top-left) and picture date Y/M/D (top-right).





## 2.2 Plant nutrients

We collected live, green samples of *A. fulva* and *C. aquatilis* at peak growing season
(July 25-August 5, 2013). Each sample consisted of 10-15 plants collected from different water
depths and multiple haphazardly selected locations in pond habitats. The collected plants were
separated into leaves and roots, then rinsed with distilled water, oven-dried at 60˚C for 24 hrs
inside open paper envelopes, then shipped to Utah State University Analytical Labs (USUAL)
for immediate processing. Most macro- and micro-nutrients in leaves of each plant were
analyzed using an inductively-coupled plasma spectrometer (ICP-MS). Nitrogen was analyzed
by combustion analysis ($HNO_3/H_2O_2$ digestion, Leco Instrument).

## 2.3 Ancillary data

Concomitant with the collection of aquatic plants for nutrient analysis, we collected soil
and water samples, harvested aboveground plant biomass, measured spectral reflectance, and
monitored most sites using time-lapse photography (Andresen et al., 2018) (Figure 2). For each
site, sediment samples from the active root soil depth of 10-20cm for each species were collected
in triplicates within the site. Samples where then combined in a plastic bag and frozen until
analysis. In the lab, soil samples were air dried for 3 days after thaw, then analyzed for physical
and chemical factors including pH, electric conductance (EC), and macronutrients (P, K, and
Nitrate). Water chemistry was determined using standard methods as described in Lougheed *et al*
(2011). In contrast to sediment, which was sampled for each plant type, water samples from open
water mid-column were assumed to be representative of the whole pond, including both plant
species given the relatively well mixed environment.
Aboveground plant biomass was harvested within duplicate representative 50cm x 20cm
quadrats for each species at each site. In addition, reflectance measurements of canopy radiance
were collected at each site employing a single channel portable spectrometer (JAZ, Ocean
Optics) following the methods of Andresen *et al* (2018). Target radiance was cross-calibrated at
every pond site using a certified 99% reflective white spectralon calibration standard (WS-1,
Labsphere), which allowed for the estimation of the reflectance ratio between plot radiance and
the calibration standard radiance. Reflectance ratio measurements were acquired with a circular
footprint of ~1 m diameter at a nadir angle from terrain. We averaged NDVI measurements from
5 scans in each plot, and 4–6 plots per pond for comparison with leaf nutrients. Normalized
Difference Vegetation Index (NDVI) was estimated from reflectance ratio values in the red and
infrared wavelengths using the formula: NDVI = (800 nm− 680 nm) / (800 nm+ 680 nm). NDVI
has become a standard proxy of plant productivity and biomass in the Arctic and has been used
to track plot (Andresen et al., 2018; Gamon et al., 2013; Soudani et al., 2012) to regional and
global seasonal and decade time-scale greening trends (Bhatt et al., 2010; Walker et al., 2012b;
Zeng and Jia, 2013). Parallel to reflectance NDVI measurements, we employed phenocams
(optical photography) at each site to calculate the "green excess" index (GEI) (Andresen et al.,
2018; Richardson et al., 2009) from peak season oblique images using the formula: [2*G - (R +



B)] where G is the brightness value in the green, R is the brightness value in the red, and B is the
brightness value in the blue. For camera details and setup refer to Andresen *et al* (2018).

### 2.4    Statistical analysis

We employed principal components analysis (PCA) to generate linear combinations of
the plant leaf nutrient data to describe the primary gradients in plant nutrient enrichment among
the sites. PCA assumes linear relationships among variables, which was confirmed with
scatterplots prior to analysis. Plant nutrient data was standardized to zero mean and unit variance
and $\log_{10}$ transformed where applicable to obtain a normal distribution. PCA axes were then
associated to environmental data (i.e. soil and water nutrients, plant biomass, NDVI, GEI) using
a Pearson correlation. Variables were log-transformed as required to meet the assumptions of
normality. All statistical analyses were performed in SAS JMP software v4.0. Significance of the
PC axes was confirmed in PC-ORD. Differences in environmental and biological characteristics
among areas within ponds dominated by *C. aquatilis* and *A. fulva* were assessed using a paired t-
test, with areas compared within each sampled pond. Green-up dates by phenocams were
determined using a regression tree analysis as described in Andresen *et al* (2018).

### 3. RESULTS

Examining the relationships between plant biomass and macronutrient (N, P) content of
the plant leaves and soil revealed that plant leaf phosphorus content was a primary determinant
of plant biomass, significantly explaining one-third of the variation in biomass for both species
(Figure 3). In addition, we found a positive linear relationship ($r = 0.7016$, $p<0.01$) between leaf
phosphorus and NDVI (Figure 3). There were no significant relationships between plant biomass
and leaf nitrogen, nor between root nutrient content and soil nutrients. Among site types,
enriched sites (Urban and Thermokarst) have statistically higher soil, leaf and water nutrients
compared to reference sites ($p<0.001$), no differences found for southern sites.
There were no significant differences in leaf, root and soil macro-nutrients among plant
species in a given pond from reference sites (paired t-test, $p>0.05$) (Table 1). However, leaf
micronutrients among plant species differed. We found significantly higher amounts of leaf Al,
B, Ba, Mn, Na, Ni, Si and Zn in *C. aquatilis* compared to *A. fulva* ($p<0.05$ level). The most
abundant leaf element in both plant species was N, followed in decreasing order by K, P, S and
Mg and these ratios were consistent across the nutrient gradient sites (Figure 4).
There were significant differences in green-up date and peak season Greenness Excess
Index (GEI) among species ($p<0.01$, Table 1). *A. fulva* greened later (day 200 vs. 183) and had
lower GEI (9 vs. 33) as compared to *C. aquatilis*. These differences are associated to unique
phenotypic properties between species in the visual spectral range (Andresen et al., 2018). There
was no corresponding difference in NDVI or biomass among species ($p>0.05$).



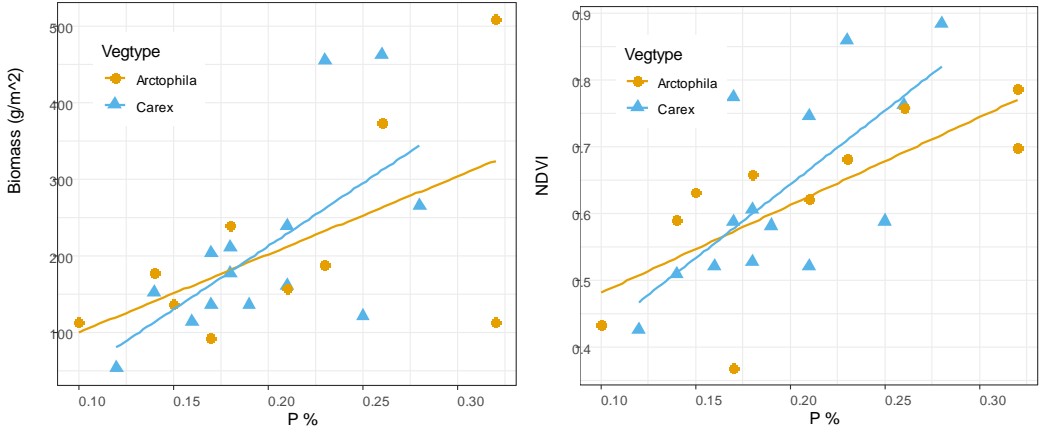


Figure 3. Relationship between Phosphorus content (%) of leaves and dry weight aboveground
biomass (left) and NDVI (right) for *Carex aquatilis* and *Arctophila fulva*. Biomass collected
during peak growing season (first week of August). Each point represents site averages. *A. fulva*
Biomass = 0.1401 + 0.0003228*P% ($R^2$=0.32, p=0.06). *C. aquatilis* Biomass = 0.1458 +
0.0002451 *P% ($R^2$=0.40, p=0.01). Both species Biomass = 0.1427 + 0.0002814*P% ($R^2$=0.34,
p=0.002). *A. fulva* NDVI = 0.1401 + 0.0003228*P% ($R^2$=0.55, p=0.01). *C. aquatilis* NDVI =
0.04996+0.2305 *P% ($R^2$=0.47, p=0.004). Both species NDVI = 0.01315+ 0.2984*P% ($R^2$=0.47,
p=0.002).

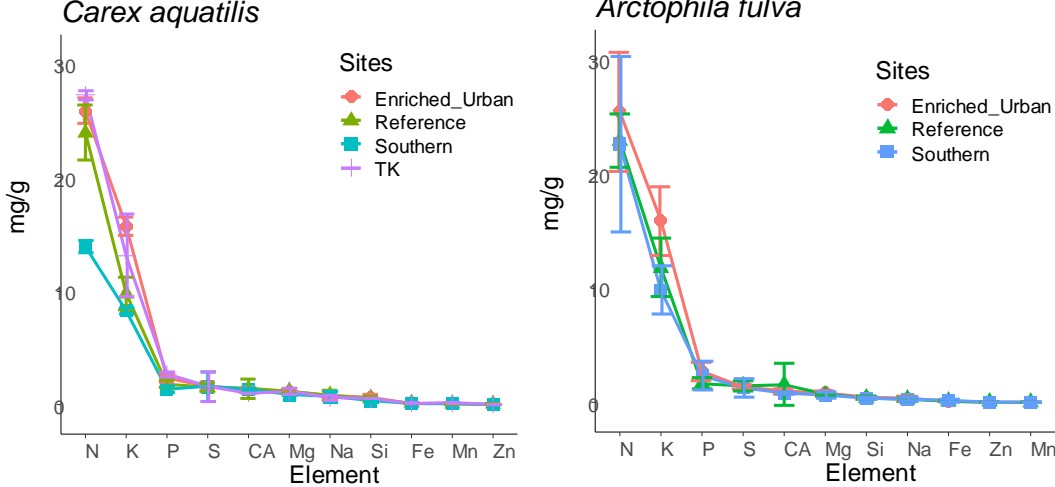


Figure 4. Descending order of element concentration in aboveground tissue among plant species.
Error bars represent one standard deviation from mean.





Table 1. Range of environmental variables by vegetation type from 17 ponds in Utqiaġvik and
Atqasuk, Alaska. (*) represents significantly different among species at p<0.01. Range
represents min and max.

| Variable | *Arctophila fulva* Mean | Range | *Carex aquatilis* Mean | Range |
|---|---|---|---|---|
| Soil pH | 5.23 | 4.7-6.3 | 5.14 | 4.7-6.3 |
| Soil EC (dS/m) | 0.86 | 0.26-2.75 | 0.589 | 0.12-2.67 |
| Soil P, available (mg/kg) | 4.78 | 2.1-10.5 | 5.625 | 2-21.3 |
| Soil K, available (mg/kg) | 42.82 | 19-80 | 44.188 | 11-109 |
| Soil Nitrate-N (mg/kg) | 1.87 | 0.01-7.6 | 1.2 | 0.01-3.8 |
| *Greening day (DOY) | 198 | 198-199 | 182 | 175-191 |
| *GEI | 8.57 | 0-18 | 33.44 | 29-37 |
| NDVI | 0.65 | 0.485-0.759 | 0.646 | 0.459-0.860 |
| Biomass (g/m$^2$) | 222.23 | 124-532 | 197.4 | 109-365 |
| Leaf TN (%) | 2.36 | 1.71-3.06 | 2.36 | 1.35-2.76 |
| Leaf P (%) | 0.2 | 0.1-0.32 | 0.2 | 0.012-0.28 |
| Root TN (%) | 1.1 | 0.67-1.45 | 0.96 | 0.69-1.2 |
| Root P (%) | 0.15 | 0.06-0.56 | 0.13 | 0.07-0.26 |


### 3.1 *Arctophila fulva*

For *A. fulva*, the first four PC axes explained 72% of the variation in plant leaf nutrients.
However, only axis 1 and 4 were significant (p<0.05). Axis 1 explained 29% of the variation and
was positively correlated with the plant macronutrients N, P, K, Ca, Mg, S as well as other
elements such as Al, B, Ba, Mn, S, Zn, and negatively correlated with Ni, Pb and Fe. On the
other hand, PC axis 4 explained 13% of the variation and was positively correlated with As, Ca,
Cr, Ni, Si, Zn. (Table 2, Figure 5).
Site types for *A. fulva* were clearly separated along axes PCA-1 and PCA-4 (Figure 5).
Enriched urban systems were located on the upper left quadrant, coinciding with higher
concentrations of many leaf nutrients and environmental variables such as soil P, EC, water P,
Si, DOC, plant biomass and higher green spectral indices (NDVI, GEI). Conversely, reference
sites and those at southern latitude were located in the opposite quadrants of the plot with a wider
distribution along PCA-4 and thus, wider variability in leaf nutrients and environmental
conditions. Southern sites for *A. fulva* showed a similar distribution to reference sites (Figure 5).



### 3.2    *Carex aquatilis*


*C. aquatilis* PC axis 1 and 2 explained 50% of the variation in the plant nutrient data. PC
axis 1 (26%) showed positive relationships with important macronutrients N, P, and Mg and
other elements such as Al, Ba, Co, Cu, Fe, Mo, Pb, Zn. PC axis 2 explained 24% of the variation
in leave nutrients and was positively associated with Al, Ba, K, Mn, P, S, Sr, and negatively
associated with Ni, Mo, Se, Zn (Table 2).
261         The *C. aquatilis* PC plot of axes 1 and 2 also showed sites grouped by type (Figure 5).
We observed a good separation along PCA-2 of enriched urban ponds as compared to reference,
southern and enriched thermokarst. Similar to *A. fulva*, the enriched sites were found at the
positive end of an axis that was positively associated with water nutrients, alkalinity,
conductance, plant biomass, NDVI and soil K (Table 3). Environmental variables positively
associated with the vertical distribution of sites along axis 1 included soil EC, water nutrients
(TDP, SRP, $NO_3$), and negative correlations with water pH, alkalinity and *C. aquatilis* green-up
date (Table 2). We noticed grouping of enriched thermokarst and reference sites for *C. aquatilis*
in a portion of the plot associated with high electrical conductance and water TDP, SRP and
$NO_3$. Conversely to *A. fulva*, the southern sites were clustered away from other sites, in the lower
left quadrant, likely reflecting earlier green up, higher GEI, and lower soil and water nutrients.

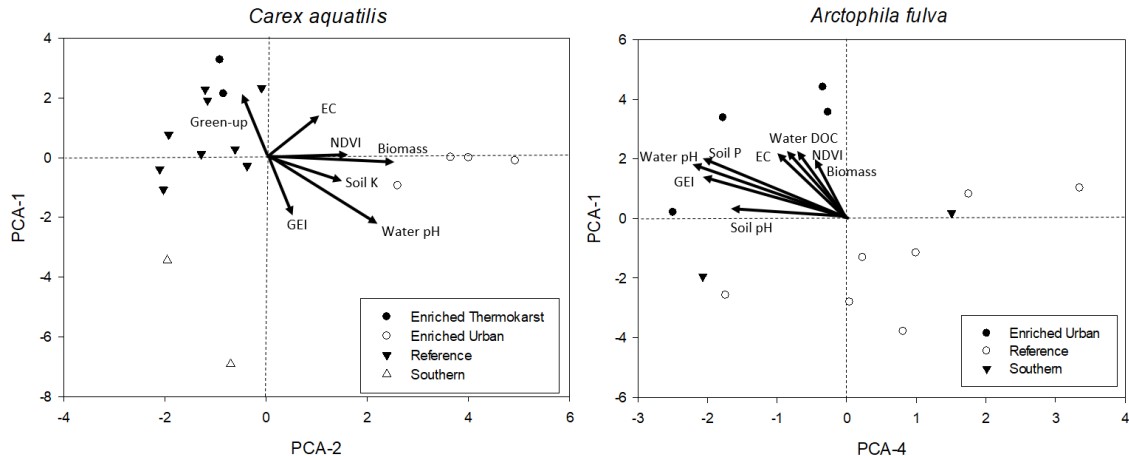

Figure 5. Plots of PCA site scores for *Arctophila fulva* (right) and *Carex aquatilis* (left) plant
nutrient data. Eigenvectors depict PCA axis correlations with environmental variables.
Eigenvectors are scaled for clarity.





Table 2. Correlation coefficients between PC axes and leaf nutrients for *Carex aquatilis* (right)
and *Arctophila fulva* (left).

| | *Carex aquatilis* | | | | | | *Arctophila fulva* | | | | |
|---|---|---|---|---|---|---|---|---|---|---|---|
| | Variance explained (%) | Axis *p-value* | Leaf Nutrient | r | p-value | | Variance explained (%) | Axis *p-value* | Leaf Nutrient | r | p-value |
| PC axis 1 | 26 | 0.001 | TN | 0.84 | 0.000 | PC axis 1 | 29 | 0.001 | P | 0.83 | 0.000 |
| | | | Cu | 0.72 | 0.001 | | | | Sr (log) | 0.81 | 0.001 |
| | | | Co (log) | 0.66 | 0.019 | | | | K | 0.80 | 0.001 |
| | | | Na (log) | 0.63 | 0.007 | | | | Al | 0.76 | 0.003 |
| | | | Mg | 0.61 | 0.009 | | | | Mg | 0.73 | 0.005 |
| | | | Pb | 0.58 | 0.016 | | | | B | 0.71 | 0.007 |
| | | | P | 0.56 | 0.019 | | | | S | 0.64 | 0.018 |
| | | | Mo | 0.54 | 0.024 | | | | Mn | 0.63 | 0.021 |
| | | | Zn | 0.53 | 0.029 | | | | Ca (log) | 0.53 | 0.061 |
| | | | Al | -0.48 | 0.051 | | | | TN | 0.50 | 0.079 |
| | | | Ba (log) | -0.73 | 0.001 | | | | Pb | -0.54 | 0.057 |
| PC axis 2 | 24 | 0.001 | S | 0.89 | 0.000 | | | | Fe (log) | -0.56 | 0.046 |
| | | | K | 0.85 | 0.000 | PC axis 4 | 13 | 0.053 | Cr | 0.86 | 0.000 |
| | | | Sr (log) | 0.74 | 0.001 | | | | As (log) | 0.80 | 0.054 |
| | | | Mn | 0.65 | 0.004 | | | | Zn | 0.58 | 0.038 |
| | | | Ba (log) | 0.59 | 0.013 | | | | Ni | 0.58 | 0.040 |
| | | | P | 0.56 | 0.020 | | | | | | |
| | | | Se (log) | -0.48 | 0.052 | | | | | | |
| | | | Ni | -0.62 | 0.008 | | | | | | |
| | | | Mo | -0.63 | 0.007 | | | | | | |
| | | | Zn | -0.66 | 0.004 | | | | | | |
| | | | Ni (log) | -0.85 | 0.000 | | | | | | |


Table 3. PC axes correlations with environmental variables.

| | *Carex aquatilis* | | | | | *Arctophila fulva* | | |
|---|---|---|---|---|---|---|---|---|
| | Environmental Variable | r | p-value | | | Environmental Variable | r | *p*-value |
| PC axis 1 | Green-up day | 0.67 | 0.049 | | PC axis 1 | Water Si | 0.84 | 0.001 |
| | Water TDP (log) | 0.56 | 0.020 | | | Water SRP (log) | 0.83 | 0.000 |
| | Water NO3 (log) | 0.52 | 0.034 | | | Water Sp. Conductance | 0.80 | 0.003 |
| | EC (log) | 0.47 | 0.069 | | | Water TDP (log) | 0.79 | 0.001 |
| | Water SRP (log) | 0.44 | 0.076 | | | Water Alkalinity | 0.78 | 0.005 |
| | Water Alkalinity (log) | -0.61 | 0.020 | | | NDVI | 0.70 | 0.008 |
| | GEI | -0.62 | 0.078 | | | Water DOC | 0.69 | 0.019 |
| | Water pH | -0.70 | 0.004 | | | Water TP (log) | 0.67 | 0.012 |
| PC axis 2 | Water Sp. Conductance (log) | 0.94 | 0.001 | | | EC (log) | 0.66 | 0.027 |
| | Water Alkalinity (log) | 0.88 | 0.001 | | | Soil P (log) | 0.61 | 0.045 |
| | Biomass | 0.84 | 0.001 | | | Biomass (log) | 0.59 | 0.034 |
| | Water pH | 0.73 | 0.002 | | | Water pH | 0.53 | 0.096 |
| | Water Si (log) | 0.58 | 0.018 | | PC axis 4 | Water pH | -0.68 | 0.021 |
| | NDVI | 0.56 | 0.071 | | | GEI | -0.67 | 0.098 |
| | Water SRP (log) | 0.54 | 0.024 | | | Soil P (log) | -0.67 | 0.025 |
| | Water TDP (log) | 0.53 | 0.029 | | | Water Alkalinity | -0.62 | 0.044 |
| | Soil K | 0.50 | 0.050 | | | Water Sp. Conductance | -0.59 | 0.057 |
| | Water TP (log) | 0.41 | 0.099 | | | Soil pH | -0.53 | 0.075 |



## 4. DISCUSSION

We explored the effects of plant nutrient enrichment in the dominant aquatic tundra species of the Arctic Coastal Plain: *A. fulva* and *C. aquatilis*. Our study is unique as it focuses on aquatic emergent plants and is based on natural responses to non-experiment, long-term nutrient enrichment compared to previous studies of fertilization treatment experiments. Plant leaf nutrients were a function of soil and water nutrients in Arctic tundra ponds. Phosphorus was the main driver of biomass in aquatic plants and plant greenness measured by NDVI in both plant species.

### 4.1 Leaf nutrients

The environmental gradient investigated in this study was highlighted by the principal component analysis and allowed better understanding of the factors influencing leaf nutrients. Our analysis shows how soil and water nutrients in ponds influence plant leaf nutrients and aboveground biomass of aquatic tundra graminoids. The Arctic is typically nutrient limited in inorganic forms of N and P in both soil (Beermann et al., 2015; Keuper et al., 2012; Mack et al., 2004) and surface waters (Rautio et al., 2011). Arctic wet sedge in particular, has been noted to be P limited given the highly organic soil which enhances recycling of N by mineralization of soil organic matter (Beermann et al., 2015; Chapin et al., 1975). Primary productivity of phytoplankton and periphyton in tundra ponds in the Utqiagvik area have been shown to be largely NP co-limited (Lougheed et al., 2015). In line with other studies in moist and wet tundra, aquatic *C. aquatilis* and *A. fulva* appear to be P limited (Beermann et al., 2015; Boelman et al., 2003; Chapin et al., 1995; Mack et al., 2004) as observed by the significant relationship between biomass and P leaf content (Figure 1). In fact, Lougheed et al (2015) suggested that macrophytes may be outcompeting algae for available nitrogen, which may account for the N limitation of algae but N sufficiency of plants. Soil nutrients were similar among cover types which may explain the homogeneous leaf macronutrient concentrations among *C. aquatilis* and *A. fulva*. However, we observed higher micronutrients and other non-essential minerals in *C. aquatilis* compared to *A. fulva*. These disparities are likely attributed to differences in taxonomic groups and thus, taxa-specific nutrient content (Chapin et al., 1975).

Compared to historical studies in the Utqiaġvik area, we found that the major plant macronutrients in *C. aquatilis* had increased since they were determined in 1970 by Chapin et al (1975). N, P and Ca plant percentage content increased from 2.18±0.09 to 2.4±0.2 (10% increase), 0.15±0.02 to 0.18±0.03 (20%), 0.08±0.02 to 0.14±0.08 (75%) respectively, for samples collected in early August. However, K and Mg were lower compared to 1970. Increase in leaf nutrients are concomitant with long-term observations of nutrient increases in tundra ponds of nitrate, ammonia and soluble reactive phosphorus (Lougheed et al., 2011). Increased plant nutrients may be a result of nutrient release from long-term increases of active layer depth (Andresen and Lougheed, 2015), thawing permafrost (Keuper et al., 2012; Reyes and Lougheed, 2015) and nitrogen mineralization (Uhlířová et al., 2007; Weintraub and Schimel, 2003) leached from terrestrial inputs. The remarkable increase in Ca observed between 1970 and 2013 is likely



associated to accumulation from high transpiration (Chapin, 1980) and suggests enhanced *C.*
*aquatilis* evapotranspiration rates compared to 50 years ago as a result of modern warmer
temperatures in both air and water (Andresen and Lougheed, 2015; Lougheed et al., 2011). It is
important to note that *C. aquatilis* has been shown to have phenotypical differences across
moisture gradients (Shaver et al. 1979). Thus, *C. aquatilis* sampled in wet meadows (Chapin et
al., 1975) might have different physiological characteristics, and therefore, different nutrient
tissue composition compared to *C. aquatilis* in aquatic habitats.

### 4.2  Nutrients, biomass, NDVI and GEI

NDVI of Arctic graminoid tundra has been noted to be a function of biomass caused by
increased nutrients (Andresen et al., 2018; Boelman et al., 2003, 2005; Epstein et al., 2012;
Raynolds et al., 2012). For example, Boelman et al. (2003) observed higher NDVI values in N
and P fertilized experimental treatments in wet sedge tundra communities compared to control
treatments. Also, Andresen et al (2018) noted higher NDVI and GEI greenness values
concomitant with higher biomass in enriched sites. Our study supports previous studies on the
importance of spectral measurements to be a function of environmental nutrient availability
through the enhancement of tundra biomass and leaf greenness at the plot level. In particular, this
study highlights phosphorus as the main nutrient augmenting aboveground biomass and plant
greenness in aquatic tundra. However, plot-scale spectral measurements such as NDVI and GEI
may differ from coarser remote sensing platforms given the spectral heterogeneity of the
radiance signal measured by the satellite sensor pixel (Guay et al., 2014) and caution should be
given to interpretations of NDVI with coarse imagery.
Increases in terrestrial productivity of the Arctic as inferred from coarse satellite NDVI
measurements have been directly attributed to increasing temperatures associated to sea ice
decline (Bhatt et al., 2010; Epstein et al., 2012). However, satellite based observations of tundra
change are complex (Myers-Smith et al., 2020) with differing trends of greening and browning
observed in recent decades (Pastick et al., 2019; Phoenix and Bjerke, 2016; Verbyla, 2008). At
the plot level, biological factors influencing spectral greenness signals include community
composition (Forbes et al., 2010) leaf area and phenology (Andresen et al., 2018; Post et al.,
2018). These factors are greatly influenced by nutrient environmental availability as shown in
this study and others (Andresen et al., 2018; Boelman et al., 2003). As permafrost degradation
and abrupt thaw events continue to increase in frequency (Andresen et al., 2020; Reyes and
Lougheed, 2015; Turetsky et al., 2020), it is imperative that we continue understanding plot-level
spectral signals and how they influence landscape-level satellite observations.
The wide range of environmental nutrient status and the broad spatial sampling
undertaken in this study provides a strong confidence on the use of spectral indices such as
NDVI to monitor environmental nutrient status at a regional scale.  In particular, the strong
relationships between NDVI and phosphorous suggest that aquatic plant communities can be
used as hot-spots and/or hot moments indicators of nutrient availability and biochemical
landscape-scale processes. Hot-spots (disproportionately high reaction rates relative to the





surrounding landscape) and hot-moments (short periods of disproportionately high reaction rates
relative to longer time periods) are generally associated with rates and reactions of biochemical
processes (e.g. nutrient cycling, productivity) and often enhanced at the terrestrial-aquatic
interface where hydrological flow-paths mobilize substrates containing complimentary reactants
(e.g. nutrients) (McClain et al., 2003). Aquatic plant communities are situated at the terrestrial-
aquatic interface inside catch-points of small landscape drainages (e.g. ponds, low-center
polygons, ice wedge pits, etc) where biogeochemical changes such as mobilization processes
from permafrost degradation (hot-moment) and nutrient mineralization (hot-moment) can be
detected and mapped (hot-spot) with spatial detail over large areas.
**5. Conclusion**
This study highlights the influence and sensitivity of aquatic tundra plant communities to
environmental nutrient status. With projected increased warming and associated terrestrial
biegeochemical processes such as increased active layer depth and permafrost thaw, increased
nutrient availability and mineralization and enhanced ecosystem carbon dynamics, aquatic plants
will continue to be a hot-spot/hot-moment of change in structure and function as they sustain
encroachment of aquatic habitats that are increasing in nutrients with potential carbon and
surface energy feedbacks to climate. Characterizing mechanisms for detection and quantification
of biogeochemical responses to climate change employing remote sensing will continue to be
pivotal into understanding spatial and temporal evolution of the Arctic terrestrial and aquatic
systems and their interactions.
**6. Appendix**
Apendix 1. Study sites and plant types. Plants species included *C. aquatilis* (C) and *A. fulva* (A).

| Site | Site type | Plant species | Latitude | Longitude |
|------|-----------|---------------|----------|-----------|
| AHMA | Enriched/urban | A,C | 71.303809 | -156.741201 |
| ATQ-E | Southern | A,C | 70.447892 | -157.362756 |
| ATQ-W | Southern | A,C | 70.457525 | -157.401083 |
| BOXER | Enriched/urban | A,C | 71.303617 | -156.752594 |
| BOXER-2 | Enriched/urban | A,C | 71.304114 | -156.748877 |
| IBP-10 | Reference | A,C | 71.2935 | -156.70433 |
| IBP-B | Reference | A,C | 71.294924 | -156.702552 |
| IBP-C | Reference | C | 71.2946 | -156.70210 |
| IBP-D | Reference | A,C | 71.294851 | -156.700166 |
| IBP-J | Reference | A,C | 71.293626 | -156.70144 |
| IBP-X | Reference | A,C | 71.295801 | -156.699817 |
| ITEX-N | Reference | A,C | 71.318141 | -156.58322 |
| TK1 | Enriched/thermokarst | C | 71.27496 | -156.632653 |
| TK3 | Enriched/thermokarst | C | 71.273975 | -156.636431 |
| UTIQ | Enriched/urban | A,C | 71.302004 | -156.722267 |
| WL02 | Reference | C | 71.2797 | -156.61891 |
| WL03 | Reference | A,C | 71.2823 | -156.61625 |





7.  **Data Availability:** Arctic data center https://arcticdata.io/

8.  **Acknowledgements:** This study was funded by the National Science Foundation (NSF) Graduate Research Fellowship Program to CGA (NSF-1110312) and research funding to VLL (ARC-0909502). Thanks to Frankie Reyes, Christina Hernandez and Nicole Miller for their help in the field. Thanks to UMIAQ, the Barrow Arctic Science Consortium (BASC) and the Ukpeagvik Inupiaq Corporation (UIC) for logistical support and land access.

9.  **Authors Contributions:** CGA and VLL collected and processed the data and wrote the manuscript.

10. **Competing interests**: No competing interests

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
