# Peer review of "Arctic aquatic graminoid tundra responses to nutrient availability"

_Biogeosciences, 2020_

## Short Comment (SC1) · 19 Nov 2020

"Arctic aquatic graminoid tundra responses to nutrient availability"

Authors: C. G. Andresen and V. L. Lougheed

This study investigates the impact of nutrient availability of emergent tundra plants, specifically the two dominant species Carex aquatilis and Arctophila fulva, in communities on the coastal plain of Alaska. This study is timely and important to the field of climate change because it investigates the role of nutrient availability on these wet tundra communities and how shifts in nutrient levels can alter ecosystem productivity.

I would like to thank the authors for the submission of this manuscript for publication and the high quality of both the study and the manuscript. The manuscript is very well

written and conclusions are sound. The study is well designed and justified. I would like to make suggestions and get clarification on just a few specific points below.

Lines 105-107: The sentence "The ACP is dominated. . .." needs citation

Lines 165-166, 172, 187, 200:For several sampling methodologies you refer to previous studies but it would be good to provide at least a sentence or two briefly describe the methods so that the reader has more of an specific idea without looking up each reference.

Lines 170-177: It would be worth noting whether there was standing water present in plots scanned with the Jaz spectrometer. If water was present did the authors make any corrections for the presence of surface water that may have altered the plot reflectance values before calculating NDVI?

Line 187: Were the phenocams facing straight down on the plots, similar to the measurement field of the Jaz? Or were they pointed out across the landscape? These are the details that would be helpful for a reader

Lines 205-206: Did the authors collect any data on the density of vegetation within each biomass and reflectance sampled plot? Some of the NDVI values seem very high and some representation of canopy and ground cover data might help explain this.

Figure 5: The legends and plots within the PCAs would be more intuitive if the shapes for each plot were consistent for each site type

Lines 325-326: It is unclear here what studies cited show a Ca increase between 1970 and 2013. It appears that the authors are citing Chapin 1980 for this point.

---

## Referee Comment (RC1) · Anonymous Referee #1 · 27 Nov 2020

**Title:** Arctic aquatic graminoid tundra responses to nutrient availability
**Author(s):** Christian G. Andresen and Vanessa L. Lougheed
**MS No.:** bg-2020-351
**MS type:** Research article

**General Comments**

Andresen and Lougheed present a well-written, interesting study on the natural responses of an aquatic plant nutrients along environmental nutrient gradient, as determined through biogeochemical methods and remote sensing-derived productivities proxies. A main strong point of this study is that it investigates the nutrient status along a terrestrial-aquatic interface, which is understudied. The main comments I have are around clarification around methodological reasoning as well as the framing of the discussion. Overall, I think this paper would be of interest to the Biogeosciences audience and adds some understanding to the role of nutrient availability in tundra plants and how these can be assessed using remote sensing (spectral signatures).

**Specific Comments**

**Abstract**

Line 21-22: Add the $r^2$ value to indicate strength of relationship

Line 25: I would suggest taking out energy balance, as that goes beyond the scope of this paper

**Introduction**

Line 37: Also López-Blanco et al. (2020), Multi-year data-model evaluation reveals the importance of nutrient availability over climate in arctic ecosystem C dynamics, ERL 15:094007.

Line 37-39: The first half of this sentence makes the previous sentence a bit redundant; I would add something about nutrient availability being the main driver of increased tundra productivity in this sentence and remove the previous one.

Line 43: I would suggest removing the reference to energy budgets and Swann et al. (2010), as that reference is largely referring to Arctic boreal shifts to deciduous cover. Here we are dealing with bare (tundra) surfaces.

Line 49-50: The latter half of this sentence (specifically "plant accumulated nutrients") is unclear.

Line 57-68: This paragraph belongs later in the introduction and could be shortened and incorporated into the concluding paragraph of the introduction.

**Methods**

Line 118-119: The authors describe four categories of sites, however Figure 1 is labelled as though there are 5 categories of sites. Consider relabelling the detailed maps using letters rather than numbers.

Line 155: I would specific that this is Total nitrogen, as opposed to just nitrate from the soil samples.

Line 164-165: Consider including the analysis method used for macronutrients. Also, what is the reasoning for selecting nitrate specifically rather also investigating total N, ammonium, and/or the organic pool? And the rationale for selecting total P rather than phosphate (as you selected the anionic form of N)?

Line 169: Is the aboveground plant biomass harvested here separate from the 10-15 plants collected for nutrient analysis?

Line 170: Include a description of the sky/solar conditions and time of day around the time of measurements

Line 170-171: What were water table conditions like at locations of reflectance measurements? How were measurements taken to ensure that they were representative of the aquatic emergent tundra without inference of water reflectance?

Line 175-183: This section is almost identical to text of another manuscript published by the authors in Andresen et al. (2018) and needs to be rewritten

Line 176-179: What is the nature of the reflectance outputs from the spectrometer employed? Did you consider averaging reflectance values from the NIR and red ranges (i.e. 62-750 nm for red) rather than using a single wavelength value?

**Results**

Line 202-204: Figure 3 does not directly back up this statement. I would move the sentence above the lack of significant relationships between plant leaf N and biomass to back this up.

Line 207: This is the first instance of root nutrient content being brought up in this paper, although it was not directly analyzed for (only collected and separate from plant leaves, according to the methods). Considering removing.

Line 240: I would put section 3.2 ahead of section 3.1, as all your figures and tables describe *C. aquatilis* before *A. fulva* (when reading left to right)

Figure 3: To help make the caption less cluttered to read, consider adding the relationships for both species with biomass and NDVI (and their respective $r^2$) to the plots, as well as adding $r_2$ values for the species relationships to the plots

Figure 5: It would be helpful for readers for ease of comparison to be consistent with symbols used for site categories. Include percent of explained variability in brackets on axes titles.

**Discussion**

There is a lot of focus on leaf nutrient status, but it would be good to see some discussion around the role of the soil nutrients and framed as a bottom-up approach (i.e. discussion of soil nutrients, the role that plays in leaf nutrient and biomass, and how that is manifested in NDVI and GEI). The latter half of this study could also benefit from discussion of study limitations, like how point-in-time measurements at peak season would differ greatly from a time-series seasonal snapshot.

Line 296-314: Much of this first paragraph talks about existing research; I would suggest the authors try to tie in more of the work from this study into this discussion. Additionally, the comparisons on nutrient limitation made here are largely to moist and wet tundra systems, however those systems can vary substantially from tundra pond environments that were studied in this manuscript. Some more justification for this is needed.

Line 315-319: It would be useful for readers to see some of this data displayed as a figure (i.e. bar graph) to visualize the changes.

Line 339-341: References for this statement? Also the statement is very generalized, as other elements not studied here have been shown to be contributing factors (i.e. growing season length, water availability, etc).

**Conclusion**

Line 377: A few sentences summarizing main findings and addressing the original research questions posed in the Introduction (line 97-99) would be helpful to tie things back together.

**Technical Corrections**

Line 33: "nutrient availability" instead of "nutrients"

Line 112: should be "on" vs "in" the edge

Line 151: "randomly" would be more appropriate than "haphazardly"

Line 290: should be "non-experimental"

---

## Author Comment (AC1) · 11 Jan 2021

**Authors response to Jeremy May comments:**

The authors will like to thank Jeremy May for his positive review and helpful comments to the manuscript. We greatly appreciate your time for providing insight.

We responded and clarified in detail to each of your comments/suggestions below (highlighted in blue)

Jeremy May commented:

*This study investigates the impact of nutrient availability of emergent tundra plants, specifically the two dominant species Carex aquatilis and Arctophila fulva, in commu- nities on the coastal plain of Alaska. This study is timely and important to the field of climate change because it investigates the role of nutrient availability on these wet tundra communities and how shifts in nutrient levels can alter ecosystem productivity.*

*I would like to thank the authors for the submission of this manuscript for publication and the high quality of both the study and the manuscript. The manuscript is very well written and conclusions are sound. The study is well designed and justified. I would like to make suggestions and get clarification on just a few specific points below.*

Lines 105-107: The sentence "The ACP is dominated. . .." needs citation

We added Hinkel et al 2003

Lines 165-166, 172, 187, 200: For several sampling methodologies you refer to previous studies but it would be good to provide at least a sentence or two briefly describe the methods so that the reader has more of an specific idea without looking up each reference.

We modify the text to provide clarity to the reader on the methods:

L165-166 as requested, we added the details on standard water chemistry methods: "*water chemistry of followed standard methods (American Public Health Association 1998) where nitrate-nitrogen was quantified by cadmium reduction; ammonia using phenate method; total phosphorus by ascorbic acid method with persulfate digestion; soluble reactive phosphorus by the ascorbic acid method; and, silica using the heteropoly blue method*".

L172- we changed "….following the methods of Andresen *et al* (2018)." To the following paragraph to read "Following Andresen et al 2018,…." for clarity given that that paragraph describes in detail Andresen et al 2018 methods already. See also next comment on lines 170-177 for full paragraph.

L187 & L200- We did not consider that these methods were critical for the manuscript and are also lengthy. No changes made.

Lines 170-177: It would be worth noting whether there was standing water present in plots scanned with the Jaz spectrometer. If water was present did the authors make any corrections for the presence of surface water that may have altered the plot reflectance values before calculating NDVI?

To clarify this, we added to lines 170-177 :

"*Following Andresen et al (2018), reflectance measurements were collected during sunny*

*conditions between 12 and 4 pm for maximum solar elevation angles ($29^0$-$33^0$, ~2pm is highest https://www.esrl.noaa.gov/) and to best match satellite observations. The person doing the collection was standing in the opposite direction of the solar azimuth angle to avoid any effects of shading by the instrument or person. All plots for both aquatic species were inundated at time of sampling (including soil, plant and spectral samples) with a water depth ($\pm SD$) of $25.2 \pm 4.6$ for A. fulva and $10.3 \pm 3.22$ cm for C. aquatilis. Solar specular reflection of water on aquatic emergent plant spectral measurements was insignificant given that solar elevation angles are relatively low in the Arctic (~$33^0$, peak season) and solar specular reflection was outside of the ~1 m spectral footprint of the measured plot.*"

Line 187: Were the phenocams facing straight down on the plots, similar to the measurement field of the Jaz? Or were they pointed out across the landscape? These are the details that would be helpful for a reader

We mention in line 185 that cameras are collecting oblique images which is a different angle to the nadir-view NDVI spectral measurements. To clarify to the reader we added in line 194: "Oblique-angle GEI collected from cameras in this study is strongly associated to nadir-angle NDVI for both *A. fulva* and *C. aquatilis* (Andresen *et al* 2018).*"*

Lines 205-206: Did the authors collect any data on the density of vegetation within each biomass and reflectance sampled plot? Some of the NDVI values seem very high and some representation of canopy and ground cover data might help explain this.

NDVI values were very high in nutrient-enriched sites due to high density of plants and their large, elongated vibrant green leaves that covered the plot . We have tiller density data for reference sites only but not for enriched. Anecdotal estimates for enriched sites are up to 10 times the density of reference sites.

Figure 5: The legends and plots within the PCAs would be more intuitive if the shapes for each plot were consistent for each site type.

We made the change as suggested

Lines 325-326: It is unclear here what studies cited show a Ca increase between 1970 and 2013. It appears that the authors are citing Chapin 1980 for this point.

We added the citation "Chapin et al (1975)" to the text for clarity

Thank you,

C.A & V. L.

---

## Author Comment (AC2) · 11 Jan 2021

Authors response to reviewer comments Title: Arctic aquatic graminoid tundra responses to nutrient availability Author(s): Christian G. Andresen and Vanessa L. Lougheed MS No.: bg-2020-351 MS type: Research article

**Dear Reviewer,**

First and foremost, the authors will like to thank you for taking the time to review in detail the manuscript and provide thoughtful and in-depth comments that made the manuscript stronger. Below, we addressed every point with a comment and the corresponding change to the manuscript text.

**General Comments from Reviewer:**

Andresen and Lougheed present a well-written, interesting study on the natural responses of an aquatic plant nutrients along environmental nutrient gradient, as determined through biogeochemical methods and remote sensing-derived productivities proxies. A main strong point of this study is that it investigates the nutrient status along a terrestrial-aquatic interface, which is understudied. The main comments I have are around clarification around methodological reasoning as well as the framing of the discussion. Overall, I think this paper would be of interest to the Biogeosciences audience and adds some understanding to the role of nutrient availability in tundra plants and how these can be assessed using remote sensing (spectral signatures).

Specific Comments from Reviewer- (Author's answers in Blue)

**Abstract**

Line 21-22: Add the  $r^2$  value to indicate strength of relationship

We added the  $R^2$  values to the abstract:

"..we identified soil phosphorus as the main limiting nutrient factor given that it was the principal driver of biomass ( $R^2=0.34$ , p=0.002) and Normalize Difference Vegetation Index (NDVI) ( $R^2=0.47$ , p=0.002) in both species."

Line 25: I would suggest taking out energy balance, as that goes beyond the scope of this paper We deleted "energy" from sentence to read: "...mobilization between terrestrial-aquatic systems and their potential influence on productivity and land-atmosphere carbon balance."

**Introduction**

Line 37: Also López-Blanco et al. (2020), Multi-year data-model evaluation reveals the importance of nutrient availability over climate in arctic ecosystem C dynamics, ERL 15:094007. Added suggested citation

Line 37-39: The first half of this sentence makes the previous sentence a bit redundant; I would add something about nutrient availability being the main driver of increased tundra productivity in this sentence and remove the previous one.

Modified the sentence to read: "Increased tundra productivity has generally been explained by warming mediated processes including increases in nutrient availability through soil warming, heterotrophic decomposition, and nutrient release from mineralization of organic matter and permafrost thaw."

Line 43: I would suggest removing the reference to energy budgets and Swann et al. (2010), as that reference is largely referring to Arctic boreal shifts to deciduous cover. Here we are dealing with bare (tundra) surfaces.

**Text changed as suggested**

Line 49-50: The latter half of this sentence (specifically "plant accumulated nutrients") is unclear. Modified text to "aboveground plant nutrients"

Line 57-68: This paragraph belongs later in the introduction and could be shortened and incorporated into the concluding paragraph of the introduction.

The authors moved this paragraph to later in the intro as suggested, after remote sensing paragraph and before the concluding paragraph in the intro.

**Methods**

Line 118-119: The authors describe four categories of sites, however Figure 1 is labelled as though there are 5 categories of sites. Consider relabelling the detailed maps using letters rather than numbers.

Changed to letters as suggested.

Line 155: I would specific that this is Total nitrogen, as opposed to just nitrate from the soil samples.

Added "Total" to sentence for clarity.

Line 164-165: Consider including the analysis method used for macronutrients. Also, what is the reasoning for selecting nitrate specifically rather also investigating total N, ammonium, and/or the organic pool? And the rationale for selecting total P rather than phosphate (as you selected the anionic form of N)?

Soil analyses only included nitrate and not TN or ammonium due to lab logistical reasons. We clarify this in the text by adding: "(For logistical reasons, only P, K, and Nitrate were analyze)"

Line 169: Is the aboveground plant biomass harvested here separate from the 10-15 plants collected for nutrient analysis?

Yes, plants for nutrient analysis were collected outside the biomass plots. No concerns emerged given that plants and densities were similar within each site.

**Line 170: Include a description of the sky/solar conditions and time of day around the time of measurements**

We added the following sentence to clarify sampling: "reflectance measurements were collected during clear sky conditions between 12 and 4 pm for maximum solar zenith angle in early August"

**Line 170-171: What were water table conditions like at locations of reflectance measurements? How were measurements taken to ensure that they were representative of the aquatic emergent tundra without inference of water reflectance?**

That's a good point that needs clarification in the methods. We clarified this by adding: "Following Andresen et al (2018), reflectance measurements were collected during sunny conditions between 12 and 4 pm for maximum solar elevation angles (290-330, ~2pm is highest https://www.esrl.noaa.gov/) and to best match satellite observations. The person doing the collection was standing in the opposite direction of the solar azimuth angle to avoid any effects of shading by the instrument or person. All plots for both aquatic species were inundated at time of sampling (including soil, plant and spectral samples) with a water depth (±SD) of 25.2 ± 4.6 for A. fulva and  $10.3 \pm 3.22$  cm for C. aquatilis. Solar specular reflection of water on aquatic emergent plant spectral measurements was insignificant given that solar elevation angles are relatively low in the Arctic (~330, peak season) and solar specular reflection was outside of the ~1 m spectral footprint of the measured plot."

Line 175-183: This section is almost identical to text of another manuscript published by the authors in Andresen et al. (2018) and needs to be rewrite

We modified the text to now read:

"The reflectance ratio was estimated between plot radiance at nadir and the calibration standard radiance. White calibration standard (38 mm wide) was positioned 30 mm at nadir below the field

spectrometer optic fiber (field of view of 25°) at each calibration, then capped closed to minimize degradation. NDVI measurements from 5 scans were averaged in each plot, and 4–6 plots per pond for comparison with leaf nutrients. Normalized Difference Vegetation Index (NDVI) was estimated from reflectance ratio values using the formula: NDVI = (800 nm- 680 nm) / (800 nm+ 680 nm). NDVI is a standard proxy of plant productivity and biomass in the Arctic and has been used to track plot (Andresen et al., 2018; Gamon et al., 2013; Soudani et al., 2012) to regional and global seasonal and decade time-scale productivity trends (Bhatt et al., 2010; Walker et al., 2012b; Zeng and Jia, 2013)."

**Original text:**

"Target radiance was cross-calibrated at every pond site using a certified 99% reflective white spectralon calibration standard (WS-1, Labsphere), which allowed for the estimation of the reflectance ratio between plot radiance and the calibration standard radiance. Reflectance ratio measurements were acquired with a circular footprint of ~1 m diameter at a nadir angle from terrain. We averaged NDVI measurements from 5 scans in each plot, and 4–6 plots per pond for comparison with leaf nutrients. Normalized Difference Vegetation Index (NDVI) was estimated from reflectance ratio values in the red and infrared wavelengths using the formula: NDVI = (800 nm- 680 nm) / (800 nm+ 680 nm). NDVI has become a standard proxy of plant productivity and biomass in the Arctic and has been used to track plot (Andresen et al., 2018; Gamon et al., 2013; Soudani et al., 2012) to regional and global seasonal and decade time-scale greening trends (Bhatt et al., 2010; Walker et al., 2012b; Zeng and Jia, 2013)."

**Line 176-179: What is the nature of the reflectance outputs from the spectrometer employed? Did you consider averaging reflectance values from the NIR and red ranges (i.e. 62-750 nm for red) rather than using a single wavelength value?**

This is a good point that also needs clarification. Reflectance outputs are pretty fair for the spectrometer. Nonetheless, we applied a moving average of 3 channels. We clarify this detail in the manuscript by adding: "We ran a moving average of 6nm (3 spectral channels) to smooth each spectral measurements and minimize noise".

**Results**

**Line 202-204: Figure 3 does not directly back up this statement. I would move the sentence above the lack of significant relationships between plant leaf N and biomass to back this up. We clarify the statement and added statistical values; text now reads:**

Examining the relationships between plant biomass and macronutrient (N, P) content of the plant leaves and soil revealed that plant leaf phosphorus content was a primary determinant of aquatic plant biomass, significantly explaining 40% of the variation in biomass of *C. aquatilis* (p=0.01) and 32% of the biomass variation of *A. fulva* (marginally significant at p=0.6). Combining both aquatic species, leaf P significantly explains 34% of aboveground biomass variability with p=0.002 (Figure 3).

Original text: Examining the relationships between plant biomass and macronutrient (N, P) content of the plant leaves and soil revealed that plant leaf phosphorus content was a primary determinant of plant biomass, significantly explaining one-third of the variation in biomass for both species (Figure 3).

Line 207: This is the first instance of root nutrient content being brought up in this paper, although it was not directly analyzed for (only collected and separate from plant leaves, according to the methods). Considering removing.

We removed "..nor between root nutrient content and soil nutrients"

Line 240: I would put section 3.2 ahead of section 3.1, as all your figures and tables describe *C*. *aquatilis*

before A. fulva (when reading left to right).

Thank you for the observation. We kept the sections as is (alphabetical order) and modified the figures and tables to read in alphabetical order left to right for consistency (A fulva on left side and C aquatilis on right side).

Figure 3: To help make the caption less cluttered to read, consider adding the relationships for both species with biomass and NDVI (and their respective  $r^2$ ) to the plots, as well as adding r2 values for the species relationships to the plots. Changed Figure 3 as suggested.

Figure 5: It would be helpful for readers for ease of comparison to be consistent with symbols used for site categories. Include percent of explained variability in brackets on axes titles. Changed Figure 5 as suggested.

**Discussion**

There is a lot of focus on leaf nutrient status, but it would be good to see some discussion around the role of the soil nutrients and framed as a bottom-up approach (i.e. discussion of soil nutrients, the role that plays in leaf nutrient and biomass, and how that is manifested in NDVI and GEI). The latter half of this study could also benefit from discussion of study limitations, like how point-in-time measurements at peak season would differ greatly from a time-series seasonal snapshot.

Seasonal variability is an interesting point that should be further investigated. Because we compared sites along a nutrient gradient during peak growing season (peak biomass and greeness), no major concerns arise about our snapshot sampling. However, we felt that it was important to note why we performed the study during peak season and acknowledge seasonal dynamics to provide a better picture to the reader. We strengthen section 4.1 and 4.2 by adding the following sentences:

a) Regarding limitations such as how point-in-time measurements at peak season would differ greatly from a time-series seasonal snapshot, we added to the discussion (Section 4.1):
"This study focused on peak season to reflect peak biomass (Andresen et al., 2017) and greenness (Andresen et al., 2018) of aquatic graminoid tundra with different environmental nutrient status. In addition, peak season is the preferred timing for assessing long-term Arctic greenness trends from satellite platforms (Bhatt et al., 2010; Walker et al., 2012a). Nutrients are known to affect seasonal phenology of aquatic graminoids by promoting earlier green-up date as well as higher season greenness (Andresen et al., 2018). However, the relationship between environmental nutrient status and seasonal plant nutrient dynamics is unclear in tundra graminoids and should be further investigated.

There are other important seasonal considerations that are worth noting. Concentrations of leaf nutrients have been shown to vary through the growing season in tundra vegetation communities. In graminoids, N and P peak within 10 days of snowmelt and gradually decrease to half of their concentration over the course of the growing season Chapin 75. On the other hand, water and soil nutrients may increase over the season in ponds as active layer thaws and soil biogeochemical processes activate (e.g. N mineralization) resulting in increased nutrient leaching from terrestrial to

aquatic systems. Evaporation and evapotranspiration likely help increase nutrient concentrations in small ponds. As climate change continues to stretch the growing season, we need to further understand seasonal dynamics of plant nutrients and its implications on productivity and land-atmosphere carbon exchange."

**b) Role of nutrients on greening (Section 4.2):**

"Aquatic tundra graminoids studied here showed higher biomass in nutrient rich sites which translated to higher plot-level greenness (e.g. NDVI, GEI). We suspect that the combination of nutrient-induced factors such as (i) increased plant density thorough increased foliage and leaf area as well as (ii) plant vitality from chlorophyll production and other pigments enhanced NDVI and GEI spectral signatures. "

Line 296-314: Much of this first paragraph talks about existing research; I would suggest the authors try to tie in more of the work from this study into this discussion. Additionally, the comparisons on nutrient limitation made here are largely to moist and wet tundra systems, however those systems can vary substantially from tundra pond environments that were studied in this manuscript. Some more justification for this is needed.

We are aware that the comparisons are to moist/wet graminoid tundra given that (to our best knowledge) there is no study on aquatic graminoid to compare to. We shorten the paragraph and added clarity to the statements.

**Paragraph now reads:**

Similar to aquatic growth forms, moist and wet tundra *C. aquatilis* and *A. fulva* appear to be P limited (Beermann et al., 2015; Boelman et al., 2003; Chapin et al., 1995; Mack et al., 2004) attributed to highly organic soil which enhances recycling of N by mineralization of soil organic matter (Beermann et al., 2015; Chapin et al., 1975). On the aquatic side, primary productivity of phytoplankton and periphyton in tundra ponds in the Utqiagvik area (including some of our sites) have been shown to be largely NP co-limited (Lougheed et al., 2015).

**Original text:**

Arctic wet sedge in particular, has been noted to be P limited given the highly organic soil which enhances recycling of N by mineralization of soil organic matter (Beermann et al., 2015; Chapin et al., 1975). Primary productivity of phytoplankton and periphyton in tundra ponds in the Utqiagvik area (including some of our study sites) have been shown to be largely NP co-limited (Lougheed et al., 2015). In line with other studies in moist and wet tundra, aquatic *C. aquatilis* and *A. fulva* appear to be P limited (Beermann et al., 2015; Boelman et al., 2003; Chapin et al., 1995; Mack et al., 2004) as observed by the significant relationship between biomass and P leaf content (Figure 1).

**Line 315-319: It would be useful for readers to see some of this data displayed as a figure (i.e. bar graph) to visualize the changes.**

We decided to keep it as text intead of a figure given that is not part of the main objective of the paper and it was ancillary information worth noting in the discussion. Otherwise, it will have to be in the results section and it will likely detract from the main take-aways of the manuscript.

Line 339-341: References for this statement? Also the statement is very generalized, as other elements not studied here have been shown to be contributing factors (i.e. growing season length, water availability, etc).

We added the appropriate references to the statement in lines 339-341:

Our study supports previous studies on the importance of spectral measurements to be a function

of environmental nutrient availability through the enhancement of tundra biomass and leaf greenness at the plot level (Andresen et al., 2018; Boelman et al., 2005).

**Conclusion**

Line 377: A few sentences summarizing main findings and addressing the original research questions posed in the Introduction (line 97-99) would be helpful to tie things back together. We improved the conclusion as suggested and tied back our research questions by adding the following sentence in the conclusion:

"In particular, we addressed that (i) aquatic graminoids were responding to higher soil and water nutrient availability through increased biomass and greenness, (ii) phosphorus was the principal limiting nutrient driving aquatic graminoid plant biomass as well as (iii) positively enhancing plotlevel NDVI spectral signatures."

**Technical Corrections**

Line 33: "nutrient availability" instead of "nutrients" Line 112: should be "on" vs "in" the edge Line 151: "randomly" would be more appropriate than "haphazardly" Line 290: should be "non-experimental"

We made the indicated corrections as suggested

Thank you, C.A. & V.L.

---

## Short Comment (SC2) · 17 Jan 2021

General Comments

This work represents an important biogeochemical study that complements our knowledge of the impact of soil fertility and lake environment on the productivity of aquatic plants in the cold Arctic climate using the example of the Barrow Peninsula, Alaska.

Specific Comments

Introduction Line 37-41: Note that, in addition to the listed cases of increase in vegetation productivity, there are many other illustrative of higher productivity of vegetation, relative to the background, within the basins of drained thermokarst lakes (Loiko et al., 2020, doi: 10.3390/plants9070867), dried up bottoms due to catastrophic events and

warm years (Nitz et al., 2020, doi: 10.5194/tc-14-4279-2020), as well as in places of activation of landslide and thermokarst processes on the slopes (Khitun et al., 2015 in Fennia - International Journal of Geography; Ukraintseva et al., 2014 in Landslides in Cold Regions in the Context of Climate Change), and sites of thawing of ground ice (Becker et al., 2016, doi: 10.1111/1365-2745.12491). The productivity of the vegetation of the listed places is high due to the higher fertility of the soils. Line 70-74: Note that there is recent information about a new unaccounted for a nutrient source that is concentrated under the active layer in the ice. These recent data make clear the reasons for the increase in productivity with increasing active layer thickness (References: Lim et al., 2020, doi: 10.1016/j.chemosphere.2020.128953 ; Subedi et al., 2020, doi: 10.5194/tc-14-4341-2020 ; Fouché et al., 2020, doi: 10.1038/s41467-020-18331-w).

Methods In this section of the article, it is also worth giving a brief description of the soils (mineral, organogenic, peat thickness). Line 150-151: At what distance from each other were the collected individual plants of the two studied species? In fact, even on a nanoscale horizontal scale, the properties of soils and sediments can change noticeably. Therefore, if the soil is not selected exactly in the place where the plant grew, then the correlation will be weaker. It is better to clarify this fact for a better understanding of the article by readers. Line 160-162: Clarify the sampling depth. Did you sample to a depth of 10-20 cm, that is, in the ranges 0-10 or 0-20? Or did you collect the horizon from 10 to 20 cm? This is important since the distribution of nutrients is highly heterogeneous in depth. The maximum concentration always falls within the 0-5 or 0-10 cm layer. However, the supply of labile forms of elements strongly depends on the soil density, which in turn depends on the type of substrate (mineral or organogenic). Line 169-170: The article indicates that the biomass was taken into account for each plant species, and not for the entire plant community. This means that in communities with several dominant species, the biomass of a particular plant species depended not only on soil fertility but also on the biomass of other plant species. Therefore, the question arises, were all communities monospecific? If there were communities of several plant species, it would be correct to normalize the biomass

to the cover (proportion of the species) of the measured species for which the biomass was measured.

Results For this section, one can calculate the ratio of nitrogen to phosphorus in the biomass of the studied plants (according to Wassen et al., doi: 10.1038/nature03950). This interesting indicator shows which of the elements limits the formation of above-ground biomass, phosphorus or nitrogen.

Discussion To Jones et al., 2012 (doi: 10.1029/2011JG001766) and Loiko et al., 2020 (doi: 10.3390/plants9070867) show that NDVI is affected by the thickness of the peat. Have you measured the thickness of the peat? Could the litter or peat have affected the biomass of plants, their projective cover and NDVI?

───────────────────────────────

---

## Editor Comment (EC1) · Michael Weintraub (Editor) · 1 Feb 2021

While only one of the seventeen potential reviewers I nominated completed a formal review, it appears that we have now received sufficient feedback with the short comments provided by Jeremy May and Sergey Loyko. Thus, I will request that Biogeosciences proceed and we can close the discussion once all author responses are posted (I would be grateful if you would provide a response to Sergey Loyko before we close the review).

Best wishes, Mike Weintraub Associate Editor
* * *

---

## Author Comment (AC3) · 8 Feb 2021

Response to Sergey Loiko

Thank you for your comments on the manuscript which raised some relevant points that helped strengthen the manuscript. We addressed every comment in detail and changes in the manuscript are highlighted them in blue below:

General Comments by Sergey Loiko:

*This work represents an important biogeochemical study that complements our knowledge of the impact of soil fertility and lake environment on the productivity of aquatic plants in the cold Arctic climate using the example of the Barrow Peninsula, Alaska.*

Specific Comments by Loiko:
Introduction Line 37-41: Note that, in addition to the listed cases of increase in vegetation productivity, there are many other illustrative of higher productivity of vegetation, relative to the background, within the basins of drained thermokarst lakes (Loiko et al., 2020, doi: 10.3390/plants9070867), dried up bottoms due to catastrophic events and warm years (Nitz et al., 2020, doi: 10.5194/tc-14-4279-2020), as well as in places of activation of landslide and thermokarst processes on the slopes (Khitun et al., 2015 in Fennia - International Journal of Geography; Ukraintseva et al., 2014 in Landslides in Cold Regions in the Context of Climate Change), and sites of thawing of ground ice (Becker et al., 2016, doi: 10.1111/1365-2745.12491). The productivity of the vegetation of the listed places is high due to the higher fertility of the soils.
To highlight the aforementioned processes, we added a follow up sentence in the introduction reading:
*In addition, abrupt thaw and recent lake drainage events enhanced during warm summers has also contributed to increased productivity through the availability of fertile soils (Turetsky et al 2020, Loiko et al 2020, Nitze et al 2020, Jones et al 2012).*

Line 70-74: Note that there is recent information about a new unaccounted for a nutrient source that is
concentrated under the active layer in the ice. These recent data make clear the reasons for the increase in productivity with increasing active layer thickness (References: Lim et al., 2020, doi: 10.1016/j.chemosphere.2020.128953 ; Subedi et al., 2020, doi: 10.5194/tc-14-4341-2020 ; Fouché et al., 2020, doi: 10.1038/s41467-020-18331-w).

We agree on the release of nutrients by thawing of permafrost. We updated the references to include Fouche et al 2020 in the original sentence:

*This phenomenon [nutrient increases in aquatic habitats] will likely become more pronounced as increasing temperatures in Arctic soils continue enhancing nitrogen mineralization (Uhlířová et al 2007, Weintraub and Schimel 2003) as well as permafrost degradation and nutrient leaching (Keuper et al 2012, Reyes and Lougheed 2015, Frey and McClelland 2009, Fouché et al 2020).*

Methods in this section of the article, it is also worth giving a brief description of the soils (mineral, organogenic, peat thickness). Line 150-151: At what distance from each other were the collected individual plants of the two studied species? In fact, even on a nanoscale horizontal scale, the properties of soils and sediments can change

noticeably. Therefore, if the soil is not selected exactly in the place where the plant grew, then the correlation will be weaker. It is better to clarify this fact for a better understanding of the article by readers.

Samples of plants were taken a few meters apart (1-4m) within the area of soil sampling. We acknowledge that soils in polygonal landscapes are highly heterogeneous given cryoturbation mixing. This helps explain the low correlation coefficients between soil and leaf nutrients. However, we designed the collection to give an overall representation of plant-soil relationships for detection using remote sensing. Highlighting these challenges will help clarify the reader about the limitations and uncertainty in these processes. Therefore, we added a section in the discussion that reads:

"We designed the sample collection to give an overall representation of plant-soil relationships for detection using remote sensing. The plant leaf samples and soil samples were not taken within the exact location, but rather, plants were collected in different areas of the monotypic stands trying to have a diverse representation of the species within each pond. Similarly, soils were collected in 3 different locations within the same area and mixed together for processing. However, given the high heterogeneity in soil properties on polygonal tundra due to cryoturbation, the relationships between soil and leaf nutrients are likely weakened and may explain the low strength of the correlation coefficients."

We also added information on the organic horizon thickness on the methods section:
"Soil organic horizon varies across the landscape due to the age of the landform (i.e. drain thaw lake basin) and cryoturbation of the soil. Nonetheless, sites are located in old and ancient drain thaw lake basins where the surface organic thickness ranges between 15 and 35cm from surface (Hinkel *et al* 2003)."

Line 160-162: Clarify the sampling depth.
Did you sample to a depth of 10-20 cm, that is, in the ranges 0-10 or 0-20? Or did you collect the horizon from 10 to 20 cm? This is important since the distribution of nutrients is highly heterogeneous in depth. The maximum concentration always falls within the 0-5 or 0-10 cm layer. However, the supply of labile forms of elements strongly depends on the soil density, which in turn depends on the type of substrate (mineral or organogenic).

Soil samples were taken at a depth range of 10-20cm given that most of the root zone for these species fell in this range. The original text reads: "For each site, sediment samples from the active root soil depth of 10-20cm for each species were collected in triplicates within the site." And we added a following sentence to the text that reads: "Soil at this depth range was a combination of mineral and organics and varied among sites and within each site. Thus, the combination of 3 soil samples in each site aided to minimize soil heterogeneity discrepancies and give an overall picture of soil conditions."

Line 169-170: The article indicates that the biomass was taken into account for each plant species, and not for the entire plant community. This means that in communities with several dominant species, the biomass of a particular

plant species depended not only on soil fertility but also on the biomass of other plant species. Therefore, the question arises, were all communities monospecific? If there were communities of several plant species, it would be correct to normalize the biomass to the cover (proportion of the species) of the measured species for which the biomass was measured.

Both plant species in this study generally grow in monotypic stands surrounding the ponds and inside the ponds (A. fulva only). We added "Monotypic" to the sentences bellow for clarity:

Line 115: These graminoids are the dominant cover in aquatic habitats, generally growing *as monotypic stands* on the edge and/or inside tundra ponds (Villarreal *et al* 2012, Andresen *et al* 2017)

Line 155: Each sample consisted of 10-15 plants collected from different water depths and multiple randomly selected locations in pond habitats *within monotypic stands of each species*. Results- For this section, one can calculate the ratio of nitrogen to phosphorus in the biomass of the studied plants (according to Wassen et al., doi: 10.1038/nature03950). This interesting indicator shows which of the elements limits the formation of aboveground biomass, phosphorus or nitrogen.

Wassen et al 2005 used fertilization experiments from literature to identify general thresholds for N & P limitations. We did not use ratios in this study given that the thresholds of N:P ratios for identifying N or P limitations are not know for aquatic tundra graminoids. Therefore, we used biomass as the main indicator of nutrient limitation which is widely accepted.

Discussion- Jones et al., 2012 (doi: 10.1029/2011JG001766) and Loiko et al., 2020 (doi: 10.3390/plants9070867) show that NDVI is affected by the thickness of the peat. Have you measured the thickness of the peat? Could the litter or peat have affected the biomass of plants, their projective cover and NDVI?

We did not measure the thickness of the peat. All our sites have relatively large amounts of peat that has been mixed with mineral soils through cryoturbation in this polygonal landscape. We included a statement  in the methods with an overall information of soil organic layer :

"Soil organic horizon varies across the landscape due to the age of the landform (i.e. drain thaw lake basin) and cryoturbation of the soil. Nonetheless, sites are located in old and ancient thaw lake basins where the surface organic thickness ranges between 15 and 35cm from surface (Hinkel *et al* 2003)."